# Rickettsial Infection in Ticks from a National Park in the Cerrado Biome, Midwestern Brazil

**DOI:** 10.3390/pathogens13010013

**Published:** 2023-12-22

**Authors:** Raquel Loren dos Reis Paludo, Warley Vieira de Freitas Paula, Lucianne Cardoso Neves, Luiza Gabriella Ferreira de Paula, Nicolas Jalowitzki de Lima, Bianca Barbara Fonseca da Silva, Brenda Gomes Pereira, Gracielle Teles Pádua, Filipe Dantas-Torres, Marcelo B. Labruna, Thiago Fernandes Martins, Jonas Sponchiado, Lucas Christian de Sousa-Paula, Wellington Hannibal, Felipe da Silva Krawczak

**Affiliations:** 1Setor de Medicina Veterinária Preventiva, Departamento de Medicina Veterinária, Escola de Veterinária e Zootecnia, Universidade Federal de Goiás—UFG, Goiânia 74690-900, Brazil; raquelloren@discente.ufg.br (R.L.d.R.P.); warleyvieira@discente.ufg.br (W.V.d.F.P.); luciannecardoso@discente.ufg.br (L.C.N.); luizadepaula@ufg.br (L.G.F.d.P.); jalowitzki@discente.ufg.br (N.J.d.L.); biancabarbarafs@gmail.com (B.B.F.d.S.); brendagomes@discente.ufg.br (B.G.P.); gracielletelespadua@discente.ufg.br (G.T.P.); 2Centro Universitário de Mineiros—UNIFIMES, Mineiros 75833-130, Brazil; 3Departamento de Imunologia, Instituto Ageu Magalhães—IAM, Fundação Oswaldo Cruz (Fiocruz), Recife 50740-465, Brazil; fdtvet@gmail.com; 4Departamento de Medicina Veterinária Preventiva e Saúde Animal, Faculdade de Medicina Veterinária e Zootecnia, Universidade de São Paulo—USP, São Paulo 05508-270, Brazil; labruna@usp.br (M.B.L.); thiagodogo@hotmail.com (T.F.M.); 5Instituto Pasteur, Área Técnica de Doenças Vinculadas a Vetores e Hospedeiros Intermediários, Secretaria de Estado da Saúde de São Paulo, São Paulo 01027-000, Brazil; 6Instituto Federal de Educação, Ciência e Tecnologia Farroupilha, Campus Alegrete, Alegrete 97541-000, Brazil; jsponchiado@yahoo.com.br; 7Tick-Pathogen Transmission Unit, Laboratory of Bacteriology, National Institute of Allergy and Infectious Diseases, Hamilton, MT 59840, USA; lcsousapaula@gmail.com; 8Laboratório de Ecologia e Biogeografia de Mamíferos, Universidade Estadual de Goiás—UEG, Quirinópolis 75860-000, Brazil; wellingtonhannibal@gmail.com

**Keywords:** *Amblyomma sculptum*, *Amblyomma triste*, *Rickettsia parkeri*, *Rickettsia tillamookensis*

## Abstract

This study was carried out from February 2020 to September 2021 in Parque Nacional das Emas (PNE), a national park located in the Cerrado biome, midwestern Brazil, as well as in surrounding rural properties. Serum and tick samples were collected from dogs, terrestrial small mammals, and humans. Ticks were also collected from the environment. Dogs were infested with *Rhipicephalus linnaei* adults, whereas small mammals were infested by immature stages of *Amblyomma* spp., *Amblyomma triste*, *Amblyomma dubitatum*, and *Amblyomma coelebs*. Ticks collected from vegetation belonged to several species of the genus *Amblyomma*, including *A. coelebs*, *A. dubitatum*, *Amblyomma naponense*, *Amblyomma sculptum*, and *A. triste*. Two *Rickettsia* species were molecularly detected in ticks: *Rickettsia parkeri* in *A. triste* from the vegetation and a *Rickettsia* sp. (designated *Rickettsia* sp. strain PNE) in *A. sculptum* and *A. triste* collected from lowland tapirs (*Tapirus terrestris*). Based on short *glt*A gene fragments, this rickettsial organism showed 99.7–100% to *Rickettsia tillamookensis*. Seroreactivity to *Rickettsia* antigens was detected in 21.9% of dogs, 15.4% of small mammals, and 23.5% of humans. The present study reveals the richness of ticks and demonstrates the circulation of rickettsial agents in one of the largest conservation units in the Cerrado biome in Brazil. To our knowledge, this is the first report of a rickettsial phylogenetically related to *R. tillamookensis* in Brazil.

## 1. Introduction

The Cerrado biome is a vast ecoregion of tropical savanna, being the second largest biome in South America and the most diverse savanna in the world. In fact, it is one of the 35 hotspots of global biodiversity, harboring a considerable richness of endemic animals and plants [1,2,3]. In Brazil, it extends over a continuous area in the central region, plus discontinuous areas in the south and north of the country [2]. Despite the importance of this biome, only 6.5% of its natural vegetation cover is currently protected [4].

Parque Nacional das Emas (PNE) is one of the most important national parks within the Cerrado biome. Around 85 species of native mammals have been recorded in the PNE and its surroundings [5,6]. This national park has been the scene of several scientific research studies on animals and their associated pathogens [7,8,9]. For instance, previous studies reported the presence of at least 10 different species of ticks infesting medium- to large-sized mammals in PNE [7,10]. Among the reported tick species, *Amblyomma sculptum* (referred to as ‘*Amblyomma cajennense*’) is of great importance, not only because of its broad host range but also because it is the primary vector of *Rickettsia rickettsii,* a spotted fever group rickettsiae (SFGR). In Brazil, *R*. *rickettsii* causes Brazilian spotted fever, the deadliest tick-borne disease of the western hemisphere [11]. Nonetheless, the circulation of SFGR in *A. sculptum* and other tick species of the PNE has never been investigated.

The PNE, its ticks, and their associated pathogens represent an interesting research model to study considering the limited human influence on this area. Indeed, studies in such a preserved area could provide novel information on the tick-borne rickettsial agents circulating in Brazil. Therefore, the present study was designed to investigate the circulation of *Rickettsia* spp. in ticks from animals, humans, and the environment in PNE. We also assessed the exposure of dogs, terrestrial small mammals, and humans to rickettsiae.

## 2. Materials and Methods

### 2.1. Study Area

PNE is in the southwestern portion of Goiás state, midwestern Brazil (Figure 1). The park is named after the large number of rheas (*Rhea americana*) that inhabit the region, which comprises an area of ~132,642 hectares. The climate type is Aw (Köppen climate classification), characterized as a tropical savanna climate with dry winter and rainy summer, with up to five months of drought. The average annual temperature is around 25 °C, ranging from 20 °C to 31 °C during the autumn–winter and spring–summer, respectively. The park is surrounded by rural properties with extensive soy and corn plantations.

Tick collection sites included trails A (18°11′38″ S and 52°52′29″ W, altitude 824 m), B (18°15′38″ S and 52°53′20″ W, altitude 779 m), C (18°16′16″ S and 52°50′35″ W, altitude 778 m), and D (17°54′06″ S and 52°59′56″ W, altitude 815 m) (Figure 1). Trails A, B, and C were established in a seasonal forest with marsh areas (seasonally flooded forest) and Cerrado *sensu stricto* (typical savanna) (Figure 1). Trail D was in an open field [12]. Tick collections from the environment and small mammal trappings were carried out at the four pre-established trails during three expeditions: (1) February 2020 (summer), (2) October 2020 (spring), and (3) September 2021 (winter). Blood samples were collected from terrestrial small mammals trapped inside the PNE and from dogs and humans in the surrounding rural areas. Ticks were sampled from dogs, terrestrial small mammals, researchers, and the environment. This study was previously approved by the Chico Mendes Institute for Biodiversity (ICMBio Permit No. 70143-1), the Institutional Animal Care and Use Committee (CEUA/UFG) of the Federal University of Goiás (protocol no. 121/19), and the Research Ethics Committee (CEP/UFG) (protocol no. 3.857.973).

### 2.2. Terrestrial Small Mammal Trapping

To capture small mammals, 126 cage-type traps (97 Sherman and 29 Tomahawk) were used with different baits (mortadella, smoked sausage, bacon, peanut butter, pumpkin, banana, and apple). Traps were installed for five consecutive nights during three collection campaigns. Additionally, three pitfall stations (with six buckets of 42.5 cm diameter and 65 cm height in each station) connected with a plastic fence (20 m long and 50 cm high) [13] were installed during the same period as the cage-type traps. The total capture effort was 1890 cage-type trap-nights and 270 fall-type trap-nights. The traps were inspected daily in the morning, and the captured animals were placed in cloth bags and properly anesthetized with 90 mg/kg of ketamine +50 mg/kg of xylazine intramuscularly to collect material (ticks and blood) [14].

Blood samples from small mammals were obtained with facial or caudal venipuncture using 1 mL syringes and 13 × 0.45 mm hypodermic needles. Each animal was marked with a numbered earring (fish and small animal tag size 1; National Band and Tag, Newport, KY, USA) and identified using different identification keys and species descriptions [15]. Some captured animals, whose species identification in the field was not possible, were euthanized, fixed, and sent for later identification by a mammal taxonomist.

### 2.3. Samples from Dogs and Humans

During the field campaigns in the surroundings of the PNE, seven farms (Figure 1) were visited, and blood samples were obtained from dogs and humans. Dogs were also examined for tick infestations on the entire body and humans on clothing, arms, legs, neck, and hands. A brief questionnaire-based interview was carried out with the dog owners to select animals 4 months or older and with free access to the forest environment.

Human blood samples were collected with intermediate cubital venipuncture and dog blood samples with jugular or cephalic venipuncture. Right after collection, the blood samples were placed into tubes without anti-coagulant and kept at room temperature (25 °C) until visible clot retraction and then centrifuged at 3000× *g* for 5 min. The obtained sera were transferred into 1.5 mL tubes and stored at –20 °C until processing.

### 2.4. Detection of Antibodies to Rickettsia spp.

Sera from dogs, rodents, marsupials, and humans were tested using an immunofluorescence assay (IFA) targeting four *Rickettsia* antigens isolated from Brazil (*R. rickettsii* strain Pampulha, *Rickettsia parkeri* strain Atlantic rainforest, *Rickettsia amblyommatis* strain Ac37, and *Rickettsia bellii* strain Mogi), as previously described [16,17,18]. Briefly, sera were diluted in two-fold increments with phosphate-buffered saline (PBS) from an initial dilution of 1:64. The slides were incubated with fluorescein isothiocyanate-labelled rabbit anti-dog IgG (Sigma, St Louis, MO, USA), goat anti-mouse IgG (Sigma), sheep anti-opossum IgG (CCZ, São Paulo, Brazil), rabbit anti-human IgG (IgG, Sigma, St. Louis, MO, USA), and rabbit anti-guinea pig IgG (Sigma) for canine, Cricetidae rodent, marsupial, human, and Echymidae rodent (*Clyomys laticeps*) sera, respectively. For each sample, the endpoint IgG titer reacting with each of the four *Rickettsia* antigens was determined. An endpoint titer at least four-fold higher for a *Rickettsia* species than those observed for the other *Rickettsia* species was considered probably homologous to the first *Rickettsia* species or a very closely related species [17,19]. On each slide, a non-reactive serum (negative control) and reactive serum (positive control) from dogs, rodents, marsupials, or humans from the other studies were tested at a 1:64 dilution [14,19,20].

### 2.5. Tick Collection and Identification

Each small mammal and dog was carefully inspected for the presence of ticks for five minutes. Host-seeking ticks were collected from the environment using the cloth dragging technique and by searching visually for ticks on the vegetation, as previously described [21,22,23].

Ticks were also collected from a ‘domesticated’ tapir (*Tapirus terrestris*) that inhabited the premises of the PNE and from an anteater (*Myrmecophaga tridactyla*) found dead on the GO-341 highway (17°39′26.49″ S and 52°55′48.49″ W) during the first expedition to the PNE, as well as from the researchers’ skin or clothing during field collections.

The ticks were removed with the aid of toothless tweezers, placed in 50 mL conical tubes containing isopropyl alcohol, and kept at room temperature until taxonomic identification in the laboratory. The ticks were identified to the species level under a stereomicroscope using descriptions and taxonomic keys [24,25,26,27]. Because of the absence of taxonomic keys for *Amblyomma* larvae from Brazil, they were identified to the genus level only [28].

### 2.6. DNA Extraction and Molecular Detection of Rickettsia

The ticks collected in this study were randomly selected and individually processed for DNA extraction using the guanidine isothiocyanate and phenol/chloroform technique [29]. In addition to ticks, blood samples collected from small terrestrial mammals and humans were also subjected to DNA extraction using the DNAeasy Blood and Tissue Kit (Qiagen, Valencia, CA, USA), following the manufacturer’s recommendations.

DNA from ticks and blood samples were tested with a TaqMan real-time qPCR assay targeting a 147 bp fragment of the rickettsial citrate synthase (*glt*A) gene [30,31]. The qPCR-positive samples were additionally tested using a panel of conventional PCR assays targeting four rickettsial genes and two intergenic regions. PCR assays were performed using primers targeting fragments of the following protein-coding genes and intergenic regions: a 401 bp fragment of the *glt*A gene; a 532 bp fragment of the *ompA* gene; a 449 bp fragment of the *atp*A gene; a fragment (unknown size) of the *cox*A gene [30,32,33,34]; 351 bp and 144 bp fragments of the intergenic regions *RC1027-xthA2* and *rpm*E-*tRN*A^fmet^, respectively [35].

Conventional PCR products (*glt*A and *omp*A genes) were purified using the Wizard^®^ SV Gel and PCR Clean-Up System (Promega, Madison, WI, USA) and sequenced using the BigDyeTM Terminator v3.1 Matrix Standards Kit (Applied Biosystems, Foster City, CA, USA) at the Núcleo de Plataformas Tecnológicas (Fiocruz PE). Sequencing reactions were carried in both directions in a 3500× L Genetic Analyzer (Applied Biosystems, Foster City, CA, USA) using the same primers as for the conventional PCR assays.

PCR-negative samples were further tested using PCR protocols targeting the *16S rDNA* gene of ticks [36] or the cytochrome b (*cytB*) gene of mammals [37,38] to validate the DNA extraction protocol. If a sample did not produce any product in these PCR assays, the sample was discarded.

### 2.7. Phylogenetic Analysis and Molecular Identification

We assembled and analyzed consensus sequences using the Sequencher^®^ v. 5.4.6 (http://www.genecodes.com (accessed on 10 October 2023)), considering a Phred quality score of ≥30. Then, consensus sequences were subjected to similarity searches against the GenBank Rickettsiales database (taxid:766) using the Basic Local Alignment Search Tool (BLASTn; http://blast.ncbi.nlm.nih.gov/Blast.cgi (accessed on 10 October 2023)). For phylogenetic reconstruction, sequences of the *glt*A gene were manually retrieved from genomes of *Rickettsia* spp. available in GenBank [39]. We used the MAFFT algorithm’s integrative refinement method FFT-NS-I for sequence alignment [40]. Dataset ends were trimmed, and a final alignment of 350 bp in length was used for phylogenetic reconstruction, as previously performed [41]. In brief, we reconstructed the *Rickettsia* phylogenetic tree using maximum likelihood inference with IQ-TREE 2 [42] with an ultrafast bootstrap (1000 replicates) for branch support. The best-fit evolutionary model was determined using ModelFinder [43], implemented in IQ-TREE, and selected based on the Bayesian Information Criterion (BIC). We utilized iTOL v.5 [44] to visualize and edit the phylogenetic tree.

## 3. Results

### 3.1. Collected Samples

Samples were collected from 41 dogs, of which 53.6% (22/41) were females and 46.3% (19/41) were males. Of these, 17% (7/41) were infested with 57 ticks, all identified as adults of *Rhipicephalus linnaei* (32 females and 25 males).

A total of 26 terrestrial small mammals belonging to 12 species (two marsupials and ten rodents) was captured. In particular, 11.5% (3/26) were didelphid marsupials (including one *Didelphis albiventris* and two *Gracilinanus agilis*), and 88.5% (23/26) were rodents belonging to the Cricetidae (one *Calomys tener*, one *Oecomys cleberi,* one *Oecomys roberti*, four *Nectomys squamipes*, six *Necromys lasiurus*, six *Oligoryzomys* cf. *mattogrossae*, one *Cerradomys scotti*, one *Cerradomys maracajuensis*, and one *Oxymycterus delator*), and Echimyidae families (one *Clyomys laticeps*). Of these, 46.1% (12/26) were females, and 53.8% (14/26) were males. Considering the three trails separately, the highest capture success was on trail C, totaling 53.8% (14/26), followed by trail A at 26.9% (7/26) and trail B at 19.2% (5/26).

Out of 28 humans enrolled in this study, 35.7% (10/28) were females, and 64.3% (18/28) were males. Of the total, 17 were blood-sampled and 11 were examined for the presence of ticks on their clothing, arms, legs, necks, and hands.

A total of 3211 ticks were sampled, of which 1.8% (57/3211) were from dogs, 8.4% (271/3211) from terrestrial small mammals, 0.5% (15/3211) from a tapir, 0.4% (13/3211) from an anteater, 0.5% (15/3211) from humans, and 88.4% (2840/3211) from the environment.

### 3.2. Serology of Animals and Humans

Among dogs, 21.9% (9/41) reacted to one or more rickettsial antigens, with 7.3% (3/41) to *R. rickettsii* (endpoint titer: 64), 17.1% (7/41) to *R. bellii* (endpoint titers: 64–512) and 4.9% (2/41) to *R. amblyommatis* (endpoint titers: 64). No dog reacted to *R. parkeri.* Four dogs (9.7%) showed a homologous reaction to *R. bellii*, whereas another two dogs (4.9%) showed inconclusive reactivities to *R. rickettsii* and *R. amblyommatis*. Concerning small mammals, 8.7% (2/23) of the rodents reacted to *R. parkeri* (endpoint titer: 128), while 66.7% (2/3) of the marsupials (one *G. agilis* and one *D. albiventris*) reacted to *R. rickettsii* (endpoint titer: 128) and *R. amblyommatis* (endpoint titer: 64), respectively. Among humans, 23.5% (4/17) reacted to at least one *Rickettsia* spp. antigen, with 17.6% (3/17) of them showing a homologous reaction to *R. bellii* (endpoint titers: 128–256) (Table 1).

### 3.3. Tick Identification

Of the 3211 ticks collected, 5.9% (189/3211) were larvae, 71.2% (2287/3211) were nymphs, and 22.9% (735/3211) were adults. The collected ticks were identified as *A. sculptum* (2200 nymphs, 286 females, 364 males), *R. linnaei* (32 females, 25 males), *Amblyomma* spp. (189 larvae), *Amblyomma coelebs* (two nymphs), *Amblyomma dubitatum* (11 nymphs), *Amblyomma naponense* (one nymph, one female), *Amblyomma parvum* (one male), and *Amblyomma triste* (73 nymphs, 16 females, 10 males) (Table 2).

Most ticks (88.5%) were found on the vegetation (trails A, B, C, and D). These included *A. sculptum* (2188 nymphs and 633 adults), *A. triste* (one nymph and 13 adults), *A. coelebs* (one nymph), *A. dubitatum* (one nymph), and *A. naponense* (one nymph and one adult).

*Amblyomma sculptum* was found on the researchers’ clothing (0.4%; 13/3211) and other animals (0.5%, 15/3211). In total 11.5% (371/3211) of the ticks were collected from animals, including 8.4% from rodents and marsupials, 1.8% from dogs, 0.9% from the anteater and tapir, and 0.5% from humans (Table 2).

Among the terrestrial small mammals captured, 57.7% (15/26) were infested with a total of 271 ticks (188 larvae and 83 nymphs) that were identified as *Amblyomma* spp. (188 larvae), *A. triste* (72 nymphs), *A. dubitatum* (ten nymphs), and *A. coelebs* (one nymph). Four species of ticks were found infesting the terrestrial small mammals, with a spy hocicudo (*O. delator*) being the most infested animal with 32 *Amblyomma* larvae and 55 *A. triste* nymphs (Table 2).

Ticks collected opportunistically from the researchers were identified as *A. sculptum* (12 nymphs, two females) and *A. parvum* (one male). Finally, the tapir was infested with *A. triste* (eight females and five males) and *A. sculptum* (two females), whereas the anteater was infested with *A. sculptum* (four females and nine males) (Table 2).

The following voucher tick specimens were deposited in the tick collection “Coleção Nacional de Carrapatos do Cerrado” (CNCC) of the Veterinary and Animal Science School, Federal University of Goiás (CNCC 041–CNCC 072): 189 *Amblyomma* spp. larvae (CNCC 045, CNCC 060–062, CNCC 064–066, CNCC 068–070, CNCC 072), 1.798 *A. sculptum* nymphs (CNCC 045, CNCC 055–059), 406 *A. sculptum* adults (CNCC 047, CNCC 050, CNCC 052–053, CNCC 055), nine *A. dubitatum* nymphs (CNCC 041, CNCC 066, CNCC 068–069), two *A. coelebs* nymphs (CNCC 042, CNCC 067), 39 *A. triste* nymphs (CNCC 063, CNCC 066, CNCC 071), 11 *A. triste* adults (CNCC 043–044, CNCC 048, CNCC 051), one *A. naponense* nymph (CNCC 046), one *A. naponense* adult (CNCC 046), one *A. parvum* adult (CNCC 049), and 24 *R. linnaei* adults (CNCC 0054).

### 3.4. Molecular Detection of Rickettsia spp.

Upon qPCR testing, all samples from small mammals and humans were negative for the *glt*A gene of rickettsiae.

The ticks (*n* = 550) that were randomly selected for molecular analysis are described in Table 3. Of these, 1.4% (8/550) were positive for the *glt*A gene and 0.2% (1/550) for the *omp*A gene. Ticks that were positive only for the *glt*A gene were collected from the tapir (*A. triste* and *A. sculptum*), while the *A. triste* adult positive for both the *glt*A and *omp*A genes was collected from the environment (Table 3).

A total of seven sequences (five *glt*A, one *omp*A, and one *rpm*E-*tRNA*^fmet^) were successfully obtained and analyzed with BLAST. *glt*A sequences presented high percent identity (range 99.7–100% with 100% coverage) with *Rickettsia tillamookensis* strain Tillamook (GenBank accession number: CP060138.2), whereas the *omp*A sequence was 100% identical to *Rickettsia parkeri* strain RS (MN114096.1). Attempts to sequence other genes (*atp*A and *cox*A) or intergenic regions (*RC1027-xthA*2 and *rpm*E-*tRNA*^fmet^) for the *glt*A-positive samples were unsuccessful (short fragments, no consensus, poor reading). However, a *rpm*E-*tRNA*^fmet^ sequence was successfully generated for the *omp*A-positive tick, and the sequence was 100% identical to *R. parkeri* isolate Sandhill Crane (CP101541.1).

The rickettsial sequences generated in the present study were deposited in GenBank under the accession numbers: OR289682, OR289684, OR289685, OR728036, and OR728037 for the *glt*A of *Rickettsia* sp. strain PNE, OR289686, and OR728038 for *omp*A and *rpm*E-*tRNA*^fmet^ of R. parkeri, respectively (Table 3).

Our phylogenetic analysis based on obtained partial *glt*A gene fragments placed all five of the *Rickettsia* sp. strain PNE in a clade containing *R. tillamookensis* within the transitional group (Figure 2).

## 4. Discussion

The present study identified seven tick species and two rickettsiae and provided serological evidence of rickettsial exposure in domestic dogs, small mammals, and humans in a national park in the Cerrado biome and surrounding farms.

All terrestrial small mammals and humans tested negative for *Rickettsia* spp. using qPCR, which was somewhat expected considering that the bacteremia due to pathogenic *Rickettsia* spp. is relatively short and of low magnitude [45]. On the other hand, the production of anti-*Rickettsia* antibodies tends to increase over time and may be long-lasting [45], which increases their usefulness for epidemiological studies. In fact, 21.9% of the dogs were seroreactive to at least one rickettsial species, with 44.4% showing homologous reactions to *R. bellii*. This rickettsia infects a wide range of tick species in the Americas, including those that parasitize dogs, such as *A. aureolatum*, *A. ovale*, and *A. sculptum* [46]. The overall seropositivity found is higher than previously reported in dogs from Goiás state [47,48] and other areas in the northeast and south regions of Brazil [49,50,51]. However, the seropositivity was lower than that detected in areas where SFGR are endemic, such as in the south and southeast regions of Brazil [18,29]. Regarding humans, 23.5% were reactive for at least one rickettsial species, the probable homologous antigen involved in the reaction also being *R. bellii*. This rickettsia is considered non-pathogenic for animals and humans, but some studies suggest that this same species could play an important role in the ecoepidemiology of SFGR impairing the maintenance and transmission of pathogenic rickettsiae, such as *R. rickettsii*, in ticks [11,52].

Concerning small mammals, 15.4% were seroreactive to at least one rickettsial species. In another national park in the Cerrado biome, 14%, 28%, and 17% of the small mammals were seroreactive to *R. rickettsii*, *R. parkeri*, and *R. amblyommatis*, respectively [53]. Higher seropositivity has been reported in the south and southeast regions of Brazil [14,51,54]. Concerning rodents, *C. maracajuensis* and *O. roberti* were previously reported only in Serranópolis (south of Goiás) and the region known as Mato Grosso de Goiás (central portion of Goiás), respectively [55]. We identified both species (*C. maracajuensis* and *O. roberti*) in our study, thus extending their known distribution range in this state. Tick species collected from small mammals were identified in the following decreasing order of frequency: *A. triste* nymphs, *A. dubitatum* nymphs, and *A. coelebs* nymph. Previous studies on small rodents in the Cerrado biome indicated a predominance of immature stages of *A. coelebs*, *A. parvum*, and *A. triste* [53,56,57].

Most ticks collected from small mammals were from trail C, which has a floodplain vegetation type, as similarly observed by other studies that also reported larger amounts of ticks in wet and flooded areas [53,58,59]. Studies carried out in other areas of the Cerrado in Brazil and Pampa in Uruguay and Argentina reported an association between *A. triste* adults and swampy areas [53,58,60,61]. Among the 10 rodent species collected, five were infested with *A. triste* immature ticks; a single spy hocicudo (*O. delator*) harbored 76.4% of the ticks collected from small mammals. This aggregation pattern of infestation has been demonstrated for *Amblyomma fuscum* on the São Lourenço punaré (*Thrichomys laurentius*) [62]. *Oxymycterus* rodents have already been reported in the Atlantic Forest biome as hosts for other species of the genus *Amblyomma* [51]. Thus, we can suggest that *O. delator* may play an important role as a host of *A. triste* larvae and nymphs in PNE. However, further studies should be conducted to confirm this hypothesis.

The reduced number of *A. dubitatum* ticks on rodents and marsupials in this study suggests that small mammals are occasional hosts [63,64]. The absence of *A. parvum* on terrestrial small mammals may reflect its low density in the study area, in contrast to another study in the Cerrado biome where immature stages of *A. parvum* were frequently collected [53,56].

Although humans do not act as primary hosts for ticks, tick infestations on humans may be very common in some areas, which may increase the risk of pathogen transmission. *Amblyomma sculptum* is associated with riparian forests close to anthropized environments, where their preferred hosts (i.e., capybaras) are commonly found. These areas are often frequented by humans in search of leisure, thus increasing the risk of tick exposure [23,65]. *Rhipicephalus linnaei* can act as a competent vector of *R. rickettsii*, and in the present study, it was found infesting dogs, which are their preferred hosts. Popularly known as the brown dog tick, *R. linnaei* is an endophilic tick that is well adapted to indoor environments, thus living in very close contact with humans. This species is distributed throughout Brazilian biomes and has been reported on humans in the states of Pernambuco, Goiás, Rio de Janeiro, Pará, Mato Grosso do Sul, and Rio Grande do Sul [66,67].

Our DNA sequencing data confirmed the circulation of *R. parkeri* and a rickettsial organism (*Rickettsia* sp. strain PNE) phylogenetically related to *R. tillamookensis* in PNE. In Brazil, *R. parkeri* has already been reported in *A. triste* ticks collected from vegetation in the state of São Paulo and from wild animals in the state of Mato Grosso do Sul [68,69]. This tick is highly competent in transmitting *R. parkeri* [61,69], the agent of a milder form of spotted fever rickettsiosis in Brazil [70,71].

This is the first published report of a rickettsial organism related to *R. tillamookensis* in Brazil infecting *A. triste* and *A. sculptum* collected from the same tapir. Attempts to obtain other sequences from other DNA targets were unsuccessful. While we found a very high identity (99.7–100%) between *Rickettsia* sp. strain PNE and *R. tillamookensis*, we only obtained a short fragment (350 bp) of the *glt*A gene, which is very conserved among some rickettsial species. For this reason, we designated this organism as *Rickettsia* sp. strain PNE until new molecular data are available to confirm the identity of this species. *R. tillamookensis* was isolated in 1976 from a pool of *Ixodes pacificus* ticks collected in 1967 in Tillamook County Oregon, USA. It was first described in 1978, but only recently formally characterized as a new species belonging to the transitional group of *Rickettsia* [72]. The isolate produced low-grade fever and mild scrotal edema after intraperitoneal injection in guinea pigs (*Cavia porcellus*), but its pathogenicity is still unknown [72,73].

## 5. Conclusions

The present study confirmed the circulation of two rickettsiae in the PNE, including the pathogenic *R. parkeri* and a rickettsial organism (*Rickettsia* sp. strain PNE) phylogenetically related to *R. tillamookensis*. The circulation of other rickettsial agents in this national park cannot be ruled out considering the typical low prevalence of *R. rickettsii* in *A. sculptum* ticks, even in endemic areas [74]. Finally, visitors to PNE should be advised to protect themselves against ticks while visiting the park to reduce the risk of exposure to rickettsial organisms, including *R. parkeri*.

## Figures and Tables

**Figure 1 pathogens-13-00013-f001:**
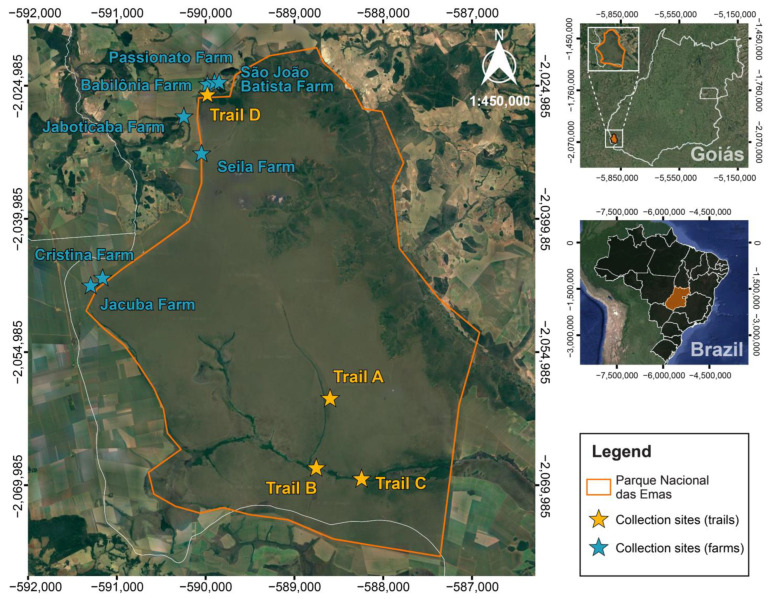
Locations of farms surrounding the park and trails within Parque Nacional das Emas (PNE) in the state of Goiás, midwestern Brazil, where animals, humans, and ticks were sampled in the present study.

**Figure 2 pathogens-13-00013-f002:**
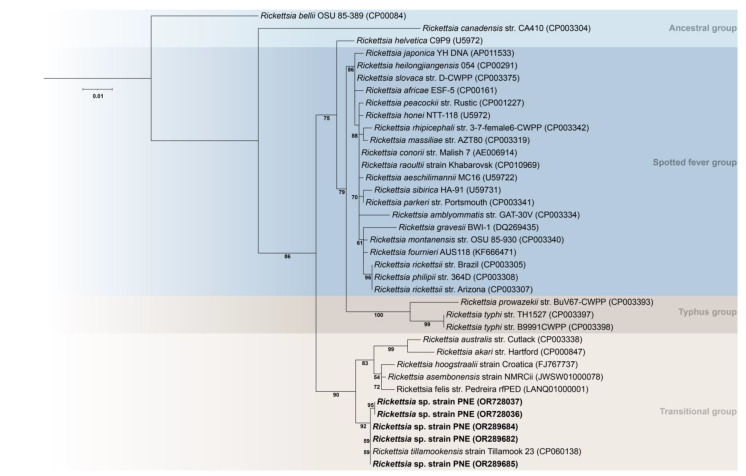
Phylogenetic tree reconstruction of the *Rickettsia* genus based on a fragment of the *glt*A gene using a dataset of 37 sequences (31 species) and 350 bp in length. The tree was inferred using the maximum-likelihood method with 1000 replicates of ultrafast bootstrap (numbers at the nodes; values < 50 were omitted) and the substitution model K3Pu + F + I. *Rickettsia bellii* OSU 85-389 (CP00084) was used as an outgroup. GenBank accession numbers are in parentheses. Sequences originally generated in the present study are in bold.

**Table 1 pathogens-13-00013-t001:** Seroreactivity to four *Rickettsia* species in humans, dogs, and terrestrial small mammals from Parque Nacional das Emas (PNE), state of Goiás, from February 2020 to September 2021.

Humans, Dogs, and Terrestrial Small Mammal Species (Tested Specimens)	Number of Animals Reactive to Each *Rickettsia* spp./% Seroreactivity (Endpoint Titers)				No. Individuals with PAIHR ^a^
	*R. rickettsii*	*R. parkeri*	*R. bellii*	*R. ammblyommatis*	
Humans (17)	0/0	0/0	3/17.6 (128–256)	1/5.9 (64)	3 *R. bellii*
Dogs (41)	3/7.3 (64)	0/0	7/17.1 (64–512)	2/4.9 (64)	4 *R. bellii*
*Calomys tener* (1)	0/0	0/0	0/0	0/0	
*Cerradomys maracajuensis* (1)	0/0	0/0	0/0	0/0	
*Cerradomys scotti* (1)	0/0	0/0	0/0	0/0	
*Clyomys laticeps* (1)	0/0	0/0	0/0	0/0	
*Didelphis albiventris* (1)	0/0	0/0	0/0	1/100 (64)	
*Gracilinanus agilis* (2)	1/50 (128)	0/0	0/0	0/0	1 *R. rickettsii*
*Necromys lasiurus* (6)	0/0	0/0	0/0	0/0	
*Nectomys squamipes* (4)	0/0	1/25 (128)	0/0	0/0	1 *R. parkeri*
*Oecomys cleberi* (1)	0/0	0/0	0/0	0/0	
*Oecomys roberti* (1)	0/0	0/0	0/0	0/0	
*Oligoryzomys* cf. *mattogrossae* (6)	0/0	0/0	0/0	0/0	
*Oxymycterus delator* (1)	0/0	1/100 (128)	0/0	0/0	1 *R. parkeri*
TOTAL (84)	4/4.8	2/2.4	10/12	4/4.8	

^a^ PAIHR, possible antigen involved in a homologous reaction. A homologous reaction was determined when the endpoint titer to a *Rickettsia* species was at least four-fold higher than the endpoint titers observed for the other three *Rickettsia* species. In this case, the *Rickettsia* species (or a very closely related genotype) involved in the highest endpoint titer was considered the PAIHR.

**Table 2 pathogens-13-00013-t002:** Ticks collected from humans and animals from Parque Nacional das Emas (PNE), state of Goiás, from February 2020 to September 2021.

Tick Host (Individuals)	A. sp.	A. coe	A. dub	A. par	A. scu		A. tri		R. lin
	L	N	N	A	N	A	N	A	A
Humans (11)				1 (1)	12 (5)	2 (1)			
Dogs (41)									57 (7)
*Didelphis albiventris* (1)	5 (1)	1 (1)	4 (1)						
*Cerradomys maracajuensis* (1)	4 (1)						1 (1)		
*Cerradomys scotti* (1)	41 (1)		1 (1)						
*Necromys lasiurus* (6)	2 (2)						3 (1)		
*Nectomys squamipes* (4)	38 (4)		5 (1)				4 (1)		
*Oligoryzomys* cf. *mattogrossae* (6)	66 (4)						9 (3)		
*Oxymycterus delator* (1)	32 (1)						55 (1)		
*Tapirus terrestris* (1)						2 (1)		13 (1)	
*Myrmecophaga tridactyla* (1)						13 (1)			
TOTAL	188 (14)	1 (1)	10 (3)	1 (1)	12 (5)	17 (3)	72 (7)	13 (1)	57 (7)

A. sp.: *Amblyomma* sp.; A. coe: *Amblyomma coelebs*; A. dub: *Amblyomma dubitatum*; A. nap: *Amblyomma naponense*; A. par: *Amblyomma parvum*; A. scu: *Amblyomma sculptum*; A. tri: *Amblyomma triste*; R. lin: *Rhipicephalus linnaei*; L: larvae; N: nymph; A: adult.

**Table 3 pathogens-13-00013-t003:** Molecular detection of rickettsial DNA in ticks collected from vegetation, humans, and animals from Parque Nacional das Emas (PNE), state of Goiás, a non-endemic area for Brazilian spotted fever, from February 2020 to September 2021.

Tick Species	Tick Stages	Source	No. Ticks with Rickettsial DNA/No. Tested Ticks (% Positive)	*Rickettsia* Species Identified by DNA Sequencing (GenBank Accession Number)
*Amblyomma sculptum*	Nymph	Vegetation	0/255 (0)	
	Adult	Vegetation	0/207 (0)	
		Human	0/1 (0)	
		*Tapirus terrestris*	1/1 (100)	*Rickettsia* sp. strain PNE (OR289682)
		*Myrmecophaga tridactyla*	0/4 (0)	
*Amblyomma triste*	Nymph	Vegetation	0/1 (0)	
		*Cerradomys maracajuensis*	0/1 (0)	
		*Necromys asiurus*	0/2 (0)	
		*Nectomys squamipes*	0/2 (0)	
		*Oligoryzomys* cf. *mattogrossae*	0/6 (0)	
		*Oxymycterus delator*	0/19 (0)	
	Adult	Vegetation	1/9 (11.1)	*R. parkeri* (OR289686, OR728038)
		*Tapirus terrestris*	6/6 (100)	*Rickettsia* sp. strain PNE (OR289684, OR289685, OR728036, OR728037)
*Amblyomma dubitatum*	Nymph	*Cerradomys scotti*	0/1 (0)	
		*Nectomys squamipes*	0/2 (0)	
		*Didelphis albiventris*	0/2 (0)	
*Rhipicephalus linnaei*	Adult	Dog	0/31 (0)	
TOTAL			8/550 (1.4)	

## Data Availability

The data presented in this study are available within this article.

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
