# Peer review of "Rickettsial Infection in Ticks from a National Park in the Cerrado Biome, Midwestern Brazil"

_pathogens, 2023, doi:10.3390/pathogens13010013_

Round 1

Reviewer 1 Report

Comments and Suggestions for Authors

I have completed the revision of the manuscript “Rickettsial infection in ticks from a National Park in the Cerrado biome, midwestern Brazil”

Excellent study, this work undoubtedly provides valuable information that contributes to the understanding of ticks and tick-borne diseases.

Some suggestions to the manuscript are referred below: 

Keywords

Organize alphabetically.

Materials and Methods

Line 77, 146... Change “5” by “five”

Results

Line 208: Necromys 5asiurus “adjust”

Author Response

Dear Reviewer 1, 

Best regards, 

Reviewer 2 Report

Comments and Suggestions for Authors

This study is well designed and the data is considerable. I think this manuscript only need some minor revisions. My comments are as below:

1. Rickettsia tillamookensis is not a spotted fever group Rickettsia, neither is R. bellii. Therefore, in the abstract line 44 " the circulation of spotted fever group rickettsiae" is not correct. Please also revise the manuscript elsewhere in the manuscript.
2. Line 132, why do you use the R. bellii? It is a phylogenetic basal Rickettsia which is not pathogenic to humans and animals.
3. Line 311, delete "of".
4. The Discussion section can be more short and concise. Meanwhile, the Introduction is too short and provides limited information. I suggest to expand it.

Comments on the Quality of English Language

The Englishi is good.

Author Response

Dear Reviewer 2, 

Besr regards, 

Reviewer 3 Report

Comments and Suggestions for Authors

I have no suggestions/comments to the methodology or design. The authors have provided a sound review of their study site and few opportunistic samplings as well, further expanding the knowledge of rickettsial field epidemiology.

Author Response

Dear Reviewer 3, 

Best regards, 
